# Spatial patterns and determinants of anxiety, depressive symptoms and their co-occurrence among currently married women of reproductive age in Bangladesh

## Research Article

anxiety; depression; mental health; spatial analysis; Bangladesh

**Corresponding author:**
Satyajit Kundu;
Email: satyajitnfs@gmail.com

Md. Aslam Hossain[1], Md. Yeasin Arafat[2] and Satyajit Kundu[3]

[1]Health Research Group, Department of Statistics, Rajshahi University, Bangladesh; [2]Department of Statistics, Islamic University, Bangladesh and [3]Public Health, School of Medicine and Dentistry, Griffith University, Australia

## Abstract

Mental health symptoms pose a significant vulnerability to stressful life events among currently married women, adversely impacting their overall well-being and quality of life. This study explores the spatial patterns and factors associated with anxiety, depressive symptoms and the co-occurrence of both symptoms among currently married women of reproductive age in Bangladesh. This study utilised data from 13,372 (weighted) currently married women aged 15–49 years in the Bangladesh Demographic and Health Survey (BDHS) 2022, which used a cross-sectional design. Multivariable logistic regression models determined the associated factors. Additionally, spatial distribution and hotspot analysis were conducted using ArcGIS version 10.8. The weighted prevalence of moderate to severe anxiety, depressive symptoms and co-occurrence of anxiety and depressive (CAD) symptoms among currently married women of reproductive age was 4.1% (95% confidence interval [CI]: 3.8%, 4.5%), 4.8% (95% CI: 4.7%, 5.4%) and 2.2% (95% CI: 2.1%, 2.6%), respectively. Clustering of anxiety symptoms (Moran's $I = 0.063$, $p < 0.001$), depressive symptoms ($I = 0.091$, $p < 0.001$) and CAD symptoms ($I = 0.082$, $p < 0.001$) were observed, with hotspots in Rangpur, Sylhet and Chittagong regions. Logistics regression analysis shows that currently married women who were living in the Barishal, Khulna, Rangpur and Sylhet regions, who belong to households with a higher wealth index, who experienced high levels of intimate partner violence (IPV), have completed high school, who are sexually inactive and whose husbands are unemployed, were more likely to experience CAD symptoms. Additionally, currently married women of reproductive age, whose age was 25–34 years, who are labourers, whose pregnancies are terminated and who have ≥5 children ever born, are at a higher risk of having anxiety symptoms. Besides, currently married women aged 25–34 years and 35–44 years, who are underweight, were more likely to have depressive symptoms. The findings highlight a significant regional disparity in the burden of anxiety, depressive and CAD symptoms among currently married women of reproductive age in Bangladesh. These findings can help design site-specific programmes and actions for women in the hot spot areas of Rangpur, Sylhet and Chittagong.

## Impact statement

This study highlights the significant mental health burden faced by currently married, reproductive-aged women in Bangladesh, emphasising the spatial patterns and socio-demographic factors contributing to anxiety, depression and their co-occurrence. By using data from the Bangladesh Demographic and Health Survey 2022, we have identified distinct regional disparities. Socio-economic factors, such as wealth, educational level, exposure to intimate partner violence, sexual inactivity and the employment status of husbands, were key determinants of mental health symptoms. The spatial autocorrelation and hot spot detection allow for a more nuanced understanding of the mental health challenges within specific regions, advocating for localised mental health interventions. The study also underscores the critical need for targeted interventions, especially in economically disadvantaged and socio-culturally marginalised communities. Furthermore, the identification of factors like domestic violence and low educational attainment as key contributors to mental health distress emphasises the importance of addressing social determinants of health in mental health strategies. This research provides evidence for the urgent need to expand mental health services in identified hotspots and to promote gender-sensitive mental health interventions that address the unique challenges faced by women.





## Introduction

Mental health distresses, particularly anxiety and depressive symptoms, have emerged as a pressing public health concern globally, contributing significantly to the overall burden of disease and disability among women of reproductive age (Ferrari et al., 2013; World Health Organization [WHO], n.d.). Mental health illnesses are a set of medical disorders that can affect feelings, mood, a person's ability to think and daily functioning (Hossain et al., 2014). Anxiety and depressive symptoms are widespread mental illness that continues to be a global problem. According to the estimates, close to one out of every eight people in the world suffers from some kind of mental illness (Ferrari et al., 2013; GBD 2019 Mental Disorders Collaborators, 2022).

According to the Diagnostic and Statistical Manual of Mental Disorders, Fifth Edition, anxiety disorders involve excessive and persistent fear or anxiety that is disproportionate to actual threats and leads to significant distress or functional impairment, not better explained by medical conditions, other mental disorders or substance use (American Psychiatric Association, 2013). Depressive disorders are defined by sustained periods of sadness, emptiness or irritability along with cognitive or physical symptoms, such as changes in sleep, appetite, energy, concentration or self-worth, that mark a clear change from prior functioning, persist for at least 2 weeks in major depressive disorder and cause clinically significant impairment (American Psychiatric Association, 2013).

Approximately 7% of individuals have anxiety, and 4% have depression, across the globe (Institute for Health Metrics and Evaluation [IHME], 2024). Predictions indicate that by 2050, depressive disorders will be the 7th cause of the global disease burden, followed by anxiety disorders in the 19th position (Vollset et al., 2024). The economic burden is equally overwhelming: depression and anxiety combined explain over US$1 trillion in lost productivity annually, a figure set to rise to US$16 trillion by 2030 (The Lancet Global Health, 2020). Depression in women elevates the risk of mortality (Ouh et al., 2025) and shortens life expectancy (Chan et al., 2023) and increases suicidality (Arnone et al., 2024).

In 2021 alone, researchers estimated that 970 million people worldwide, including 236 million in South Asia, suffered from a mental illness (GBD 2019 Mental Disorders Collaborators, 2022). The prevalence of common mental disorders among women of reproductive age in India and Myanmar was 5.8% and 21.2%, respectively (Aye et al., 2020; Annajigowda et al., 2023). Additionally, the prevalence of depressive symptoms among women of reproductive age in Nepal and Pakistan was 6% and 33.3%, respectively (Shehzad et al., 2016; Pandey et al., 2024).

For Bangladesh, a South Asian lower-middle-income nation, the modelled global estimates of anxiety and depressive disorders stand at 5% and 6%, respectively (IHME, 2024). However, more focused contextual evidence suggests these figures are likely far greater. As an example, a cross-sectional study reported that in the context of the COVID-19 pandemic, 27% of Bangladeshi female university students had anxiety, and 36% had depressive symptoms (Nahar et al., 2022). National data also shows a difference by gender in mental health, with women reporting higher rates of anxiety and depression (18.8% vs. 12.5%) than men (Hasan et al., 2021).

Depression is connected with several influential factors, including lower educational levels, poor socio-economic status, larger family sizes (Abdullah et al., 2024), lower age group >30 years (Kranjac et al., 2025) and women with more children (Golovina et al., 2023). Several key determinants of reproductive women's anxiety were self-reported health status, experienced emotional violence, pregnancy termination, husband's alcohol consumption, genital discharge (Tohan et al., 2025), lower wealth status (Noor et al., 2025), higher education level, somatic symptoms and sleep quality (Wang et al., 2023). Besides, different socio-cultural factors, such as traditional beliefs, social and religious taboos, the poor availability of mental health care services, low mental health literacy, social stigma and capital, adversely influence mental healthcare-seeking behaviours among women in Bangladesh (Dutta et al., 2022; Koly et al., 2022). Women in Bangladesh often exhibit inadequate knowledge about mental health disorders, management and their prevention (WHO, 2020).

Mental health is not only a private requirement but also a global human right necessary for achieving sustainable development, especially in low- and middle-income countries (LMICs) (Patel et al., 2018). Prior studies in Bangladesh were conducted on the determinants of depression and anxiety. However, they focused on garment workers, the effect of coronavirus disease 2019 (COVID-19), university students, women living in urban slums and construction workers (Fitch et al., 2017; Haque et al., 2022; Islam et al., 2020; Koly et al., 2023; Roy et al., 2024). A recent study using Bangladesh Demographic and Health Survey (BDHS) 2022 data revealed that 4% of women experienced moderate to severe anxiety, and 5% had moderate to severe depression (Raza et al., 2025). The prevalence was notably higher in the Khulna, Rangpur and Sylhet divisions. Key risk factors included older age and greater household decision-making autonomy (Raza et al., 2025). Despite the mental health burden, care-seeking remained low; only 22% of symptomatic women consulted a provider, and just 8% received medication (Raza et al., 2025). However, the above studies did not explore spatial patterns in symptom distribution, leaving a critical gap in understanding geographic disparities in mental health outcomes. Also, global studies often vary due to differences in modelling methods and data quality, underscoring further the need for context-specific spatial analyses using local data.

A study conducted in Peru reported that the expected prevalence of depressive symptoms was higher in the southern departments of Peru (Villarreal-Zegarra et al., 2025). The study emphasised the importance of identifying hotspots of depression and evaluating the distribution of socio-economic inequality across the country (Villarreal-Zegarra et al., 2025). Hotspots are defined as clusters of high data values, while cold spots are clusters of low data values. Hot and cold spots can help detect areas with high and low prevalence of anxiety, depressive symptoms and their co-occurrence among reproductive women in specific regions of Bangladesh (Rima et al., 2025). This geographical insight is crucial for designing focused strategies and policies to address and reduce geographical disparities in mental health burden (Gruebner et al., 2011; Ulrich et al., 2023). Hence, pinpointing geographic areas where women experience a high burden of mental health distress using geographic information systems is essential for developing location and context-specific interventions. Therefore, this study aims to assess the spatial patterns of anxiety, depressive symptoms and their co-occurrence, and identify the factors associated with these mental health outcomes among currently married women of reproductive age in Bangladesh. We hypothesised that mental health symptoms are clustered geographically, and that socio-demographic, health and reproductive factors significantly influence their likelihood of experiencing these symptoms.

Thus, this study was guided by the following research questions (RQ):

**RQ1:** What are the spatial patterns of anxiety, depressive symptoms and their co-occurrence among currently married women of reproductive age in Bangladesh?

**RQ2:** What are the factors associated with anxiety, depressive symptoms and their co-occurrence among currently married women of reproductive age in Bangladesh?

## Methods

### Study settings, data source, study design and participants

Bangladesh is located in the northeastern part of South Asia, which is one of the most densely populated countries. It comprises 64 districts along with eight administrative divisions: Dhaka, Mymensingh, Barishal, Khulna, Sylhet, Rajshahi, Rangpur and Chattogram. This research leverages the analysis of secondary data from a cross-sectional survey from the 2022 BDHS. The BDHS, a nationwide survey conducted every few years, employs a two-stage stratified cluster sampling method. During the first stage, enumeration areas (EAs) are selected, followed by the selection of households within each chosen EA in the second stage. This approach ensures the sample is representative of the entire Bangladeshi population. The BDHS targeted households across rural and urban EAs, aiming to complete interviews with married women aged 15–49 years. This study utilised the women's individual record file (National Institute of Population Research and Training and ICF, 2023). In BDHS 2022, a total of 30,078 households were selected, including eligible married women aged 15–49 years for interviews. During data management for this study, we removed the cases based on the exclusion criteria from both the outcome and all explanatory variables. After applying these exclusions, data from a weighted sample of 13,372 currently married women of reproductive age (15–49 years) were included in the final analysis.

### Exclusion criteria

In general, mental health symptoms were more common among divorced and widowed women (Trivedi et al., 2009; Mokhtari et al., 2013). Besides, mothers are more likely to experience mental health issues, especially if their children had diarrhoea, fever or cough in the past 2 weeks (Jiang et al., 2021). For this reason, 16,640 women were omitted from the analysis out of 30,078 reproductive-aged married women, based on the following criteria: 10,090 with missing data; 2,336 not living with their husbands at the time of the survey; and 174 separated, 532 widowed, and 260 divorced. Additionally, 393, 2,242 and 613 children were affected by diarrhoea, fever and cough, respectively, in the last 2 weeks. These exclusions helped align with our research focus on currently married women of reproductive age. After these exclusions, a total of 13,372 (weighted) currently married women of reproductive age were included in the final analysis.

### Outcome variables

The primary outcome variables were symptoms of anxiety and depression. Anxiety was measured using the Generalised Anxiety Disorder Scale (GAD-7), which is a 7-item self-reported tool to assess the occurrence and intensity of generalised anxiety symptoms over the previous 2 weeks. Women's responses to each item were recorded on a 4-point Likert scale from 0 (*never*) to 3 (*always*), resulting in total scores ranging from 0 to 21 (Spitzer et al., 2006). GAD-7 score is categorised as follows: 0–4 indicates none, 5–9 mild, 10–14 moderate and 15–21 severe anxiety symptoms. A score of 10 or above was categorised as having symptoms of moderate to severe anxiety (Spitzer et al., 2006). The GAD-7 scale demonstrated strong reliability with a Cronbach's $\alpha$ of 0.834.

Depressive symptoms were assessed using the Patient Health Questionnaire (PHQ-9), a self-reported instrument comprising nine items evaluating depressive symptoms over the last 2 weeks (Kroenke et al., 2001). Similar to anxiety, women's responses to each item were recorded on a 4-point Likert scale, resulting in total scores ranging from 0 to 27, delineating the following severity levels: no depression (scores 0–4), mild (scores 5–9), moderate (scores 10–14), moderately severe (scores 15–19) and severe (scores 20–27). A score of 10 or above was categorised as symptoms of having moderate to severe depression (Kroenke et al., 2001; Banik et al., 2022). The PHQ-9 scale also showed strong internal consistency with a Cronbach's $\alpha$ of 0.821. Previous studies in Bangladesh also used the same cut-off score of 10 on both the PHQ-9 and GAD-7 to assess the presence of depressive and anxiety symptoms, respectively (Banik et al., 2022; Wahid et al., 2023).

### Explanatory variables

We selected explanatory variables that we hypothesised could be associated with anxiety, depressive symptoms and their co-occurrence, including geographical region (Barishal, Chattogram, Dhaka, Khulna, Mymensingh, Rajshahi, Rangpur and Sylhet); mass media exposure (no and yes), wealth index (poorest, poorer, middle, richer and richest), household members (in numbers) ($\geq 4$ and $\geq 5$), intimate partner violence (IPV) (low and high), women age (years) (15–24, 25–34, 35–44 and $\geq 45$), education level (less than high school, high school and more than high school), occupation (homemakers, business, service holder and labourer), body mass index (BMI) (underweight, normal weight and overweight), sexual inactivity (no and yes), pregnancy termination (no and yes), children ever born (1–2, 3–4 and $\geq 5$), husband's current age (in years) (<40, 40–50, 50–60 and >60), husband's occupation (unemployed, farmer/labourer, business and service holder).

Mass media exposure was constructed from three variables: television viewing, radio listening and newspaper reading. It was categorised as "yes" if a woman had exposure to any of these media sources or "no" if she had no exposure to any of them (Fentie et al., 2023). The wealth index, constructed by the Demographic and Health Survey (DHS) programme, is a composite measure of a household's cumulative living standard, calculated using principal component analysis (PCA) (Rutstein, 2015). IPV is assessed using five pieces of information: If the wife is subjected to violence by her husband for any of the following reasons: (i) going out without telling the husband, (ii) neglecting the children, (iii) arguing with the husband, (iv) refusing to have sex with the husband and (v) burning the food. Using PCA, we have two categories of IPV as low and high (M. Islam et al., 2021; Rahman et al., 2022).

BMI was calculated as weight (kg)/height (m$^2$). BMI categories were defined according to World Health Organization classifications: underweight (<18.5 kg/m$^2$), normal (18.5–24.9 kg/m$^2$) and overweight ($\geq 25.0$ kg/m$^2$) (Khanam et al., 2021). Sexual inactivity was measured as the frequency of sexual intercourse among women in the last month. The married women in their reproductive age were asked, "About how many times did you have sex during the

last month?" If women replied that no sexual activity occurred during the last month, it was considered sexual inactivity (Laksono et al., 2020). Most of the exposure variables and their categories have been identified from the source of prior studies (Maria et al., 2021; Abdullah et al., 2024; Tohan et al., 2025).

### Statistical analysis

The analysis of data was conducted using two software packages: (i) STATA version 17.0, developed by StataCorp in College Station, Texas and (ii) ArcGIS version 10.8. The weighted prevalence of anxiety, depressive symptoms and their co-occurrence was calculated by considering respective weights for the unit of analysis. To account for the complex survey design and sampling weights, the "svyset" command was utilised in STATA. The "svy" command was consequently used to obtain valid statistical inferences. Data cleaning procedures were conducted before analysis, including handling missing values and recoding variables as necessary (Croft et al., 2020). Descriptive statistics were generated to assess the prevalence of anxiety, depressive symptoms and their co-occurrence among currently married reproductive-aged women. Categorical variables were analysed using Pearson's chi-square tests (bivariate analysis) to compare the prevalence of anxiety, depressive symptoms and their co-occurrence.

Multivariable logistic regression analysis was performed to obtain the factors associated with the outcome variables (i.e., anxiety symptoms, depressive symptoms and co-occurrence of anxiety and depressive [CAD] symptoms). Only the significant variables in bivariate analysis were entered into the regression models. Subsequently, unadjusted and adjusted odds ratios (ORs) with 95% confidence intervals (CIs) were calculated using the logistic regression models. The final model was assessed for multicollinearity, and no multicollinearity was detected. Goodness-of-fit of the adjusted regression models was measured using the Hosmer and Lemeshow test. Hosmer and Lemeshow test estimates indicate that the regression models for anxiety ($\chi^2$ = 1.02, $p$ = 0.4187), depressive symptoms ($\chi^2$ = 1.04, $p$ = 0.4045) and their co-occurrence ($\chi^2$ = 1.24, $p$ = 0.2797) are a good fit because the insignificant $p$-value suggests that there is no considerable difference between observed and predicted values. All statistical tests were two-sided, with a significance level set at $p$-values < 0.05.

### Spatial autocorrelation and hot spot analysis

Global spatial autocorrelation (Global Moran's I) statistic was conducted to examine the spatial patterns of anxiety, depressive symptoms and their co-occurrence among currently married reproductive-aged women using ArcGIS version 10.8. The weighted prevalence of anxiety, depressive symptoms and their co-occurrence was calculated for each cluster using STATA. This cluster-wise prevalence was then merged with the cluster number and geographic coordinate data (point shapefile) of each cluster using ArcGIS software. For the BDHS 2022, global positioning system data of 674 clusters were found among 675 clusters included in the survey. One cluster did not have any location data, and the final study identified data from a total of 674 clusters. For spatial analysis, the refined dataset was transferred to Excel before being imported into the ArcGIS 10.8 software. This analysis helped to determine whether the high prevalence of anxiety, depressive and CAD symptoms was dispersed, clustered or randomly distributed in Bangladesh.

Positive Moran's I value (close to +1) indicates geographic clustering for anxiety, depressive symptoms and their co-occurrence prevalence is high; the negative Moran's value (close to −1) indicates geographic clustering for anxiety, depressive symptoms and their co-occurrence prevalence is low, while Moran's I value 0 indicates random distribution. A statistically significant Moran's value ($p$ < 0.05) had a chance to reject the null hypothesis, suggesting the presence of spatial autocorrelation (Chen, 2013). Hot spot analysis using the Getis-Ord Gi* Statistic calculates $z$-scores and $p$-values. These $z$-scores and $p$-values indicate whether the observed spatial clustering of high or low values is more pronounced than would be expected in a random distribution (Environmental Systems Research Institute, n.d.). Areas with statistically significantly high $z$-scores represent clusters of high prevalence (hotspots), and significantly low $z$-scores, including negative values (cold spots), represent clusters of low prevalence of anxiety, depressive symptoms and their co-occurrence among currently married reproductive-aged women (Villarreal-Zegarra et al., 2025).

## Results

### Prevalence of anxiety, depressive symptoms and co-occurrence of both symptoms

The weighted prevalence among currently married reproductive-aged women of moderate to severe anxiety symptoms, depressive symptoms and CAD symptoms was 4.1% (95% CI: 3.8%, 4.5%), 4.8% (95% CI: 4.7%, 5.4%) and 2.2% (95% CI: 2.1%, 2.6%), respectively. Women living in the Rangpur division (5.3%), those who experienced high levels of IPV (5.3%), women aged 45 years or older (6.1%), those with less than a high school education (4.6%), labourers (around 7%), sexually inactive women (6.6%), women who had terminated pregnancies (5.5%) and women with five or more children (8.1%) faced higher rates of anxiety symptoms. Women whose husbands are 60 years or older (7.4%) and those with unemployed husbands (8.4%) are also associated with increased anxiety symptoms. Additionally, women who living in the Rangpur region (7.9%), those who experienced high levels of IPV (7.1%), women aged 45 years or older (6.8%), those with less than a high school education (4.9%), who were underweight (7.4%), sexually inactive women (7.2%) and women with five or more children (6.7%) faced higher rates of depressive symptoms. Women whose husbands are 60 years or older (6.7%) and those with unemployed husbands (8.8%) are also associated with increased depressive symptoms (Supplementary Table 1).

Besides, 3.4% of currently married reproductive-aged women living in the Sylhet region experienced CAD symptoms, while the poorest households account for 2.6% of cases. Around 3% of women who experienced high levels of IPV face CAD symptoms. Married women of reproductive age who were 45 years or older (3.9%), those with less than a high school education (2.6%), sexually inactive women (3.4%), women who have had pregnancy terminations (2.7%) and those with five or more children ever born (4.0%) have higher rates of CAD symptoms. CAD symptoms are also more common among women whose husbands are 60 years or older (4.3%) and among women whose husbands are unemployed (6.2%) (Supplementary Table 1).

### Associated factors of anxiety symptoms

Table 1 presents the associated factors of anxiety symptoms among Bangladesh's currently married reproductive-aged women.

**Table 1.** Factors associated with anxiety symptoms among currently married reproductive-aged women in Bangladesh

| Study variable | Anxiety symptoms | | | |
| | UOR (95% CI) | *p*-value | AOR (95% CI) | *p*-value |
| --- | --- | --- | --- | --- |
| **Geographical region** | | | | |
| Barishal | 1.41 (0.86–2.31) | 0.168 | **1.84 (1.05–3.19)** | **0.031** |
| Chattogram | 1.48 (0.88–2.49) | 0.142 | 1.64 (0.91–2.96) | 0.096 |
| Dhaka | 1.20 (0.77–1.87) | 0.418 | 1.32 (0.77–2.25) | 0.308 |
| Khulna | 1.89 (1.22–2.93) | 0.005 | **2.41 (1.44–4.04)** | **0.001** |
| Mymensingh | Reference | | Reference | |
| Rajshahi | 0.89 (0.53–1.46) | 0.631 | 0.85 (0.45–1.58) | 0.605 |
| Rangpur | 1.79 (1.15–2.82) | 0.011 | **2.19 (1.29–3.69)** | **0.003** |
| Sylhet | 1.68 (1.05–2.70) | 0.031 | **2.19 (1.29–3.72)** | **0.003** |
| **Intimate partner violence** | | | | |
| Low | Reference | | Reference | |
| High | 1.42 (1.13–1.77) | 0.002 | 1.27 (0.95–1.70) | 0.104 |
| **Age (years)** | | | | |
| 15–24 | Reference | | Reference | |
| 25–34 | 2.04 (1.45–2.87) | <0.001 | **1.73 (1.09–2.74)** | **0.020** |
| 35–44 | 2.79 (1.99–3.89) | <0.001 | 1.51 (0.86–2.66) | 0.150 |
| ≥45 | 3.46 (2.37–5.04) | <0.001 | 1.35 (0.69–2.66) | 0.373 |
| **Education level** | | | | |
| Less than high school | 2.13 (1.44–3.14) | <0.001 | 1.61 (0.99–2.63) | 0.057 |
| High school | 1.64 (1.09–2.48) | 0.019 | 1.61 (0.99–2.62) | 0.054 |
| More than high school | Reference | | Reference | |
| **Occupation** | | | | |
| Homemakers | Reference | | Reference | |
| Business | 1.35 (0.97–1.88) | 0.072 | 1.32 (0.88–1.97) | 0.175 |
| Service holder | 0.55 (0.25–1.18) | 0.126 | 0.61 (0.23–1.61) | 0.318 |
| Labourer | 1.79 (1.24–2.61) | 0.002 | **1.63 (1.01–2.64)** | **0.047** |
| **Sexual inactivity** | | | | |
| No | Reference | | Reference | |
| Yes | 1.81 (1.30–2.52) | <0.001 | **1.54 (1.06–2.24)** | **0.023** |
| **Pregnancy termination** | | | | |
| No | Reference | | Reference | |
| Yes | 1.55 (1.27–1.89) | <0.001 | **1.30 (1.03–1.65)** | **0.028** |
| **Children ever born** | | | | |
| 1–2 | Reference | | Reference | |
| 3–4 | 1.52 (1.24–1.87) | <0.001 | 1.04 (0.80–1.34) | 0.780 |
| ≥5 | 2.65 (1.90–3.69) | <0.001 | **1.63 (1.04–2.54)** | **0.032** |
| **Husband's current age (years)** | | | | |
| <40 | 0.55 (0.43–0.71) | <0.001 | **0.51 (0.32–0.81)** | **0.005** |
| 40–49 | Reference | | Reference | |
| 50–59 | 1.25 (0.97–1.61) | 0.078 | 0.74 (0.52–1.05) | 0.088 |
| ≥60 | 1.70 (1.25–2.33) | 0.001 | 1.22 (0.75–1.96) | 0.421 |

(*Continued*)

**Table 1.** (Continued)

| Study variable | Anxiety symptoms | | | |
| --- | --- | --- | --- | --- |
| | UOR (95% CI) | *p*-value | AOR (95% CI) | *p*-value |
| **Husband's occupation** | | | | |
| Unemployed | 3.27 (1.91–5.60) | <0.001 | 2.06 (0.96–4.41) | 0.061 |
| Farmer/labourer | 1.52 (0.96–2.41) | 0.072 | 1.30 (0.71–2.38) | 0.385 |
| Business | 1.40 (0.88–2.25) | 0.158 | 1.25 (0.68–2.31) | 0.467 |
| Service holder | Reference | | Reference | |
| Hosmer and Lemeshow test | $\chi^2$ value = 1.02; *P*-value = 0.4187 | | | |

*Note*: The bolded estimates are statistically significant. χ2, chi-square; AOR, adjusted odds ratio; CI, confidence interval; UOR, unadjusted odds ratio.

Multivariable logistic regression revealed that women in Khulna (adjusted OR [AOR]: 2.41; 95% CI: 1.44–4.04), Rangpur (AOR: 2.19; 95% CI: 1.29–3.69), Sylhet (AOR: 2.19; 95% CI: 1.29–3.72) and Barishal (AOR: 1.84; 95% CI: 1.05–3.19) divisions were significantly more likely to experience anxiety compared with women in Mymensingh. Women aged 25–34 years were 1.73 times more likely (AOR: 1.73; 95% CI: 1.09–2.74) to have anxiety symptoms compared to women aged 15–24 years. Anxiety symptoms were strongly associated with education; women with high school or below high school education had increased odds (AOR: 1.61; 95% CI: 0.99–2.63 and AOR: 1.61; 95% CI: 0.99–2.62, respectively), although only marginally significant. Women working as labourers were significantly more likely to have anxiety symptoms (AOR: 1.63; 95% CI: 1.01–2.64) than homemakers. Sexually inactive females were 1.54 times more likely to experience anxiety symptoms (AOR: 1.54; 95% CI: 1.06–2.24) than their sexually active peers. Individuals whose past involved pregnancy termination also had increased odds of having anxiety symptoms (AOR: 1.30; 95% CI: 1.03–1.65). Five or more children were more likely to experience anxiety symptoms (AOR: 1.63; 95% CI: 1.04–2.54) than one or two. Additionally, those women were less likely to have anxiety symptoms, whose husbands were under the age of 40 years (AOR: 0.51; 95% CI: 0.32–0.81) than those having husbands aged 40–49 years old.

### Associated factors of depressive symptoms

Table 2 shows associated factors of depressive symptoms among Bangladesh's currently married reproductive-aged women. Women residing within the Barishal region had a higher experience of having depressive symptoms compared to women residing in Mymensingh (AOR: 2.04; 95% CI: 1.03–4.07). The odds remained higher for women residing in Khulna (AOR: 2.96; 95% CI: 1.57–5.60) and Rangpur (AOR: 2.35; 95% CI: 1.12–4.92), indicating strong regional disparities in depressive symptoms. Women exposed to high levels of IPV were 1.69 times more likely to report depressive symptoms compared with the low exposure group (AOR: 1.69; 95% CI: 1.13–2.54). Age was also a significant factor; women aged 25–34 years (AOR: 2.19; 95% CI: 1.30–3.68) and 35–44 years (AOR: 2.08; 95% CI: 1.08–4.01) were more likely to report depressive symptoms than the lowest age group (15–24 years). Underweight women had increased odds of experiencing depressive symptoms compared to women with normal BMI (AOR: 1.69; 95% CI: 1.02–2.83). Sexual inactivity was also significant, in that sexually inactive women had over twice the likelihood of

experiencing depressive symptoms compared to sexually active women (AOR: 2.16; 95% CI: 1.33–3.49). Besides, women having unemployed husbands had almost three times higher odds of having depressive symptoms compared to women having husbands who were service holders (AOR: 2.99; 95% CI: 1.08–8.29).

### Associated factors of the CAD symptoms

Table 3 shows the associated factors influencing the CAD symptoms in currently married reproductive-aged women in Bangladesh. There was a significant difference based on region, with women residing in Khulna (AOR: 4.14; 95% CI: 1.92–8.91), Sylhet (AOR: 3.98; 95% CI: 1.86–8.53), Rangpur (AOR: 2.81; 95% CI: 1.21–6.53) and Barishal (AOR: 2.68; 95% CI: 1.25–5.75) showing higher odds of having comorbid depressive and anxiety symptoms compared to those living in Mymensingh. In terms of household wealth status, "Richer" women were 1.99 times more likely (AOR: 1.99; 95% CI: 1.12–3.53) to experience CAD symptoms compared to women from the "Richest" category. Experiencing high levels of IPV was another important factor (AOR: 1.47; 95% CI: 1.01–2.15), which increased the odds of having comorbid mental symptoms. Women with less than a high school education (AOR: 2.49; 95% CI: 1.22–5.08) and high school education (AOR: 2.42; 95% CI: 1.27–4.59) were more likely to report comorbid depressive and anxiety symptoms compared to those with more than a high school education. Sexual inactivity was also associated with higher odds of comorbid mental health symptoms (AOR: 1.69; 95% CI: 1.00–2.84). Finally, women who had unemployed husbands had higher chances of experiencing CAD symptoms (AOR: 2.40; 95% CI: 1.05–5.49) than those having husbands who were service holders.

### Spatial distribution of anxiety, depressive symptoms and their co-occurrence

Three maps were created to illustrate the spatial distribution of currently married reproductive-aged women's anxiety, depressive symptoms and their co-occurrence (Supplementary Figures S1–S3). The global spatial autocorrelation analysis revealed that the distribution of currently married women's anxiety, depressive symptoms and their co-occurrence across Bangladesh was not random. Instead, it showed clustered patterns of anxiety symptoms (global Moran's $I$ = 0.063, $p$ < 0.001), depressive symptoms (global Moran's $I$ = 0.091, $p$ < 0.001) and CAD symptoms (global Moran's $I$ = 0.082, $p$ < 0.001; Figure 1).

**Table 2.** Factors associated with depressive symptoms among currently married reproductive-aged women in Bangladesh

| Study variable | Depressive symptoms | | | |
|---|---|---|---|---|
| | UOR (95% CI) | *p*-value | AOR (95% CI) | *p*-value |
| **Geographical region** | | | | |
| Barishal | 1.59 (1.03–2.48) | 0.037 | **2.04 (1.03–4.07)** | **0.042** |
| Chattogram | 1.17 (0.69–1.99) | 0.553 | 1.60 (0.76–3.38) | 0.213 |
| Dhaka | 1.20 (0.76–1.91) | 0.429 | 1.66 (0.83–3.31) | 0.149 |
| Khulna | 2.09 (1.39–3.15) | <0.001 | **2.96 (1.57–5.60)** | **0.001** |
| Mymensingh | Reference | | Reference | |
| Rajshahi | 0.97 (0.62–1.53) | 0.915 | 1.09 (0.54–2.20) | 0.806 |
| Rangpur | 2.54 (1.66–3.89) | <0.001 | **2.35 (1.12–4.92)** | **0.023** |
| Sylhet | 1.97 (1.23–3.17) | 0.005 | 2.01 (0.94–4.30) | 0.072 |
| **Intimate partner violence** | | | | |
| Low | Reference | | Reference | |
| High | 1.70 (1.36–2.13) | <0.001 | **1.69 (1.13–2.54)** | **0.010** |
| **Age (years)** | | | | |
| 15–24 | Reference | | Reference | |
| 25–34 | 1.65 (1.26–2.17) | <0.001 | **2.19 (1.30–3.68)** | **0.003** |
| 35–44 | 1.68 (1.28–2.22) | <0.001 | **2.08 (1.08–4.01)** | **0.028** |
| ≥45 | 2.31 (1.69–3.16) | <0.001 | 2.02 (0.86–4.75) | 0.105 |
| **Education level** | | | | |
| Less than high school | 1.64 (1.17–2.30) | 0.004 | 1.49 (0.87–2.57) | 0.145 |
| High school | 1.48 (1.09–2.01) | 0.010 | 1.52 (0.93–2.49) | 0.093 |
| More than high school | Reference | | Reference | |
| **Body mass index** | | | | |
| Underweight | 1.63 (1.13–2.36) | 0.010 | **1.69 (1.02–2.83)** | **0.041** |
| Normal weight | Reference | | Reference | |
| Overweight | 1.05 (0.81–1.35) | 0.714 | 1.05 (0.76–1.45) | 0.742 |
| **Sexual inactivity** | | | | |
| No | Reference | | Reference | |
| Yes | 1.64 (1.23–2.19) | 0.001 | **2.16 (1.33–3.49)** | **0.002** |
| **Children ever born** | | | | |
| 1–2 | Reference | | Reference | |
| 3–4 | 1.48 (1.20–1.82) | <0.001 | 1.13 (0.79–1.59) | 0.492 |
| ≥5 | 1.70 (1.25–2.32) | 0.001 | 0.98 (0.47–2.04) | 0.965 |
| **Husband's current age (years)** | | | | |
| <40 | 0.82 (0.66–1.02) | 0.073 | 1.26 (0.84–1.89) | 0.262 |
| 40–49 | Reference | | Reference | |
| 50–59 | 1.26 (0.99–1.61) | 0.058 | 1.26 (0.74–2.16) | 0.397 |
| ≥60 | 1.43 (1.02–2.00) | 0.036 | 1.00 (0.46–2.18) | 0.992 |
| **Husband's occupation** | | | | |
| Unemployed | 2.95 (1.78–4.91) | <0.001 | **2.99 (1.08–8.29)** | **0.035** |
| Farmer/labourer | 1.48 (1.00–2.19) | 0.050 | 1.22 (0.63–2.34) | 0.556 |
| Business | 1.61 (1.06–2.44) | 0.025 | 1.42 (0.74–2.69) | 0.288 |
| Service holder | Reference | | Reference | |
| Hosmer and Lemeshow test | $\chi^2$ value = 1.04; *P*-value = 0.4045 | | | |

*Note*: The bolded estimates are statistically significant. χ2, chi-square; AOR, adjusted odds ratio; CI, confidence interval; UOR, unadjusted odds ratio.

**Table 3.** Factors associated with the CAD symptoms among currently married reproductive-aged women in Bangladesh

| Study variable | CAD symptoms | | | |
| --- | --- | --- | --- | --- |
| | UOR (95% CI) | *p*-value | AOR (95% CI) | *p*-value |
| **Geographical region** | | | | |
| Barishal | 1.96 (1.06–3.61) | 0.031 | **2.68 (1.25–5.75)** | **0.012** |
| Chattogram | 1.84 (0.87–3.89) | 0.109 | 2.23 (0.89–5.52) | 0.084 |
| Dhaka | 1.49 (0.78–2.85) | 0.227 | 1.88 (0.82–4.28) | 0.134 |
| Khulna | 2.79 (1.51–5.17) | 0.001 | **4.14 (1.92–8.91)** | **<0.001** |
| Mymensingh | Reference | | Reference | |
| Rajshahi | 1.32 (0.67–2.59) | 0.425 | 1.08 (0.43–2.72) | 0.861 |
| Rangpur | 2.25 (1.17–4.31) | 0.015 | **2.81 (1.21–6.53)** | **0.017** |
| Sylhet | 2.90 (1.56–5.38) | 0.001 | **3.98 (1.86–8.53)** | **<0.001** |
| **Wealth Index** | | | | |
| Poorest | 1.78 (1.12–2.83) | 0.015 | 1.73 (0.93–3.20) | 0.081 |
| Poorer | 1.24 (0.76–2.04) | 0.389 | 0.95 (0.51–1.77) | 0.877 |
| Middle | 1.78 (1.08–2.94) | 0.024 | 1.63 (0.85–3.12) | 0.137 |
| Richer | 1.83 (1.14–2.94) | 0.012 | **1.99 (1.12–3.53)** | **0.019** |
| Richest | Reference | | Reference | |
| **Intimate partner violence** | | | | |
| Low | Reference | | Reference | |
| High | 1.46 (1.08–1.97) | 0.013 | **1.47 (1.01–2.15)** | **0.046** |
| **Age (years)** | | | | |
| 15–24 | Reference | | Reference | |
| 25–34 | 2.15 (1.39–3.31) | 0.001 | 1.65 (0.93–2.93) | 0.083 |
| 35–44 | 2.45 (1.56–3.84) | <0.001 | 1.27 (0.63–2.59) | 0.502 |
| ≥45 | 4.03 (2.51–6.46) | <0.001 | 1.68 (0.75–3.74) | 0.205 |
| **Education level** | | | | |
| Less than high school | 3.09 (1.77–5.43) | <0.001 | **2.49 (1.22–5.08)** | **0.012** |
| High school | 2.32 (1.32–4.05) | <0.001 | **2.42 (1.27–4.59)** | **0.007** |
| More than high school | Reference | | Reference | |
| **Sexual inactivity** | | | | |
| No | Reference | | Reference | |
| Yes | 1.72 (1.14–2.59) | 0.009 | **1.69 (1.00–2.84)** | **0.049** |
| **Pregnancy termination** | | | | |
| No | Reference | | Reference | |
| Yes | 1.33 (1.01–1.76) | 0.045 | 1.25 (0.89–1.77) | 0.193 |
| **Children ever born** | | | | |
| 1–2 | Reference | | Reference | |
| 3–4 | 1.68 (1.29–2.19) | <0.001 | 1.11 (0.79–1.55) | 0.534 |
| ≥5 | 2.39 (1.61–3.53) | <0.001 | 1.37 (0.69–2.73) | 0.363 |
| **Husband's current age (years)** | | | | |
| <40 | 0.53 (0.39–0.73) | <0.001 | 0.67 (0.43–1.04) | 0.077 |
| 40–49 | Reference | | Reference | |
| 50–59 | 1.06 (0.79–1.44) | 0.680 | 1.04 (0.67–1.61) | 0.852 |
| ≥60 | 1.68 (1.12–2.53) | 0.012 | 1.11 (0.57–2.14) | 0.757 |

(*Continued*)

**Table 3.** (*Continued*)

| Study variable | CAD symptoms | | | |
| --- | --- | --- | --- | --- |
| | UOR (95% CI) | *p*-value | AOR (95% CI) | *p*-value |
| **Husband's occupation** | | | | |
| Unemployed | 3.68 (2.02–6.69) | <0.001 | **2.40 (1.05–5.49)** | **0.038** |
| Farmer/labourer | 1.14 (0.68–1.92) | 0.617 | 0.89 (0.43–1.84) | 0.752 |
| Business | 1.23 (0.72–2.13) | 0.447 | 1.04 (0.53–2.06) | 0.905 |
| Service holder | Reference | | Reference | |
| Hosmer and Lemeshow test | $\chi^2$ value = 1.24; *P*-value = 0.2797 | | | |

*Note*: The bolded estimates are statistically significant. χ2, chi-square; AOR, adjusted odds ratio; CAD, co-occurrence of anxiety and depressive symptoms; CI, confidence interval; UOR, unadjusted odds ratio.

### Hot spot analysis of anxiety, depressive symptoms and their co-occurrence

The hotspots for higher prevalence of anxiety, depressive symptoms and their co-occurrence were identified using red colour, and while blue areas (cold spot areas) indicate divisions with a low percentage of mental health symptoms, as shown in the maps. From Figure 2, hotspot areas for the anxiety symptoms were identified in Rangpur, Sylhet and Chittagong divisions (Figure 2a), hotspot areas for the depressive symptoms were identified in Rangpur and Sylhet divisions (Figure 2b), and in Sylhet and Chittagong divisions, hotspot areas for the CAD symptoms among currently married reproductive-aged women were identified (Figure 2c). Whereas in Mymensingh and Rajshahi divisions, cold spot areas for the anxiety symptoms were detected, and the cold spots for depressive and CAD symptoms were identified in Chittagong and Mymensingh divisions (Figure 2).

### Discussion

In Bangladesh, the overall prevalence of moderate to severe anxiety, depressive symptoms and CAD symptoms was 4.1%, 4.8% and 2.2%, respectively, among currently married women of reproductive age during the 2022 survey year. Our findings align with the Global Burden of Disease (GBD) 2021 estimates, which report that among reproductive-aged women in Bangladesh, the prevalence of anxiety disorders is 6% (95% CI: 4–8%) and depressive disorders is 7% (95% CI: 5–9%) (IHME, 2024). A previous study in Bangladesh revealed that the prevalence of depression among married women in the Rajshahi district was 21.2% (Wadood et al., 2023). Sparling et al. (2020) show that 20% of reproductive-aged women of the Sylhet region in rural areas of Bangladesh screened positive for major depression. A systematic review of 24 studies conducted during COVID-19 in Bangladesh showed a pooled prevalence of 47% for both anxiety and depression (Hosen et al., 2021).

The differences between our assessments and those from other Bangladeshi studies may be due to various factors, including differences in study design, measurement tools (e.g., screening instruments used), threshold scores for defining anxiety or depression, population characteristics, cultural perceptions of mental health and sample sizes. A study in Nepal reported that the prevalence of anxiety and depressive symptoms among women was 8% and 6%, respectively (Pandey et al., 2024). In India, anxiety and depressive symptoms were most common among reproductive-aged women, with a prevalence of 4% and 3%, respectively (Annajigowda et al., 2023). People living in South Asian countries like Bangladesh,

where mental health is deeply rooted in the social, spiritual, cultural, historical, religious and holistic aspects of human lives, share all these psychosocial constructs to varying extents (Rama et al., 2014; Gopalkrishnan, 2018). Bangladesh still has critically low human resources for mental health diagnosis, treatment and prevention and to address mental health problems across communities (Trivedi et al., 2007; Thara and Padmavati, 2013). For these reasons, mental health estimates in Bangladesh differ from other LMICs despite similar contexts.

The study found that women in the wealthiest households were significantly less likely to have CAD symptoms than their richer counterparts. Specifically, women in the "richer" wealth category had nearly twice as high a chance of having co-occurring symptoms as women in the "richest" category. This finding is supported by earlier studies that report lower socio-economic status as a strong and stable predictor of adverse mental health outcomes in low- and middle-income nations (Lund et al., 2010; Patel et al., 2018). Socio-economic deprivation may increase psychological vulnerability along different pathways, such as chronic stress, limited access to high-quality healthcare and exclusion from society (Reiss, 2013). In the South Asian context, poor women often have multiple overlapping stressors, for example, food insecurity, fewer rights in decision-making and exposure to domestic stress, extending their vulnerability to the risk of common mental disorders (Anik et al., 2023). Moreover, intersections between wealth inequality, cultural stigma surrounding mental illness and climate-related stressors, such as flooding and heat waves, can intensify psychological distress by compounding social exclusion and limiting adaptive coping strategies (Ayeb-Karlsson, 2020; Charlson et al., 2021).

The results of our study indicated that significant association between exposure to IPV and the prevalence of depressive and anxiety symptoms among currently married women of reproductive age. This is consistent with previous research that IPV, including emotional and physical abuse, is a key predictor of adverse mental health consequences in women (Garcia-Moreno et al., 2006; Devries et al., 2013). Exposure to IPV can lead to long-term stress, emotional abuse and helplessness, all of which play a role in causing mental illnesses such as depression and anxiety (Afifi et al., 2009). Another study conducted in Bangladesh also found that women with a history of domestic violence were more likely to experience psychological distress, as well as symptoms of depression and anxiety (M. J. Islam et al., 2017). Moreover, socio-economic disadvantage reflected by a lower wealth index has been linked to higher IPV risk, while cultural stigma surrounding marital conflict and climate-related stressors, such as floods or heatwaves, further exacerbate women's mental health vulnerabilities in Bangladesh

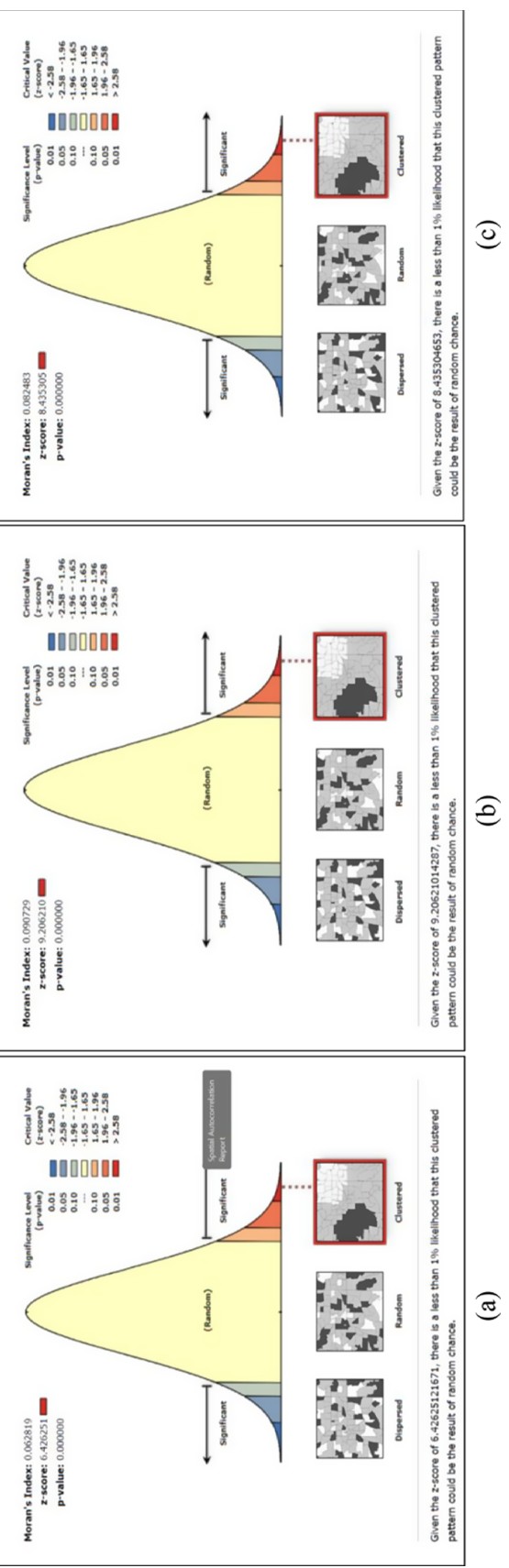

**Figure 1.** (a) Global spatial autocorrelation report of the anxiety symptoms. (b) Global spatial autocorrelation report of the depressive symptoms. (c) Global spatial autocorrelation report of CAD symptoms. All maps were generated using ArcGIS v10.8 software (https://www.arcgis.com/index.html) with data from the BDHS 2022 survey, utilising the base shapefile of Bangladesh from a freely available online source: https://data.humdata.org/dataset/cod-ab-bgd?

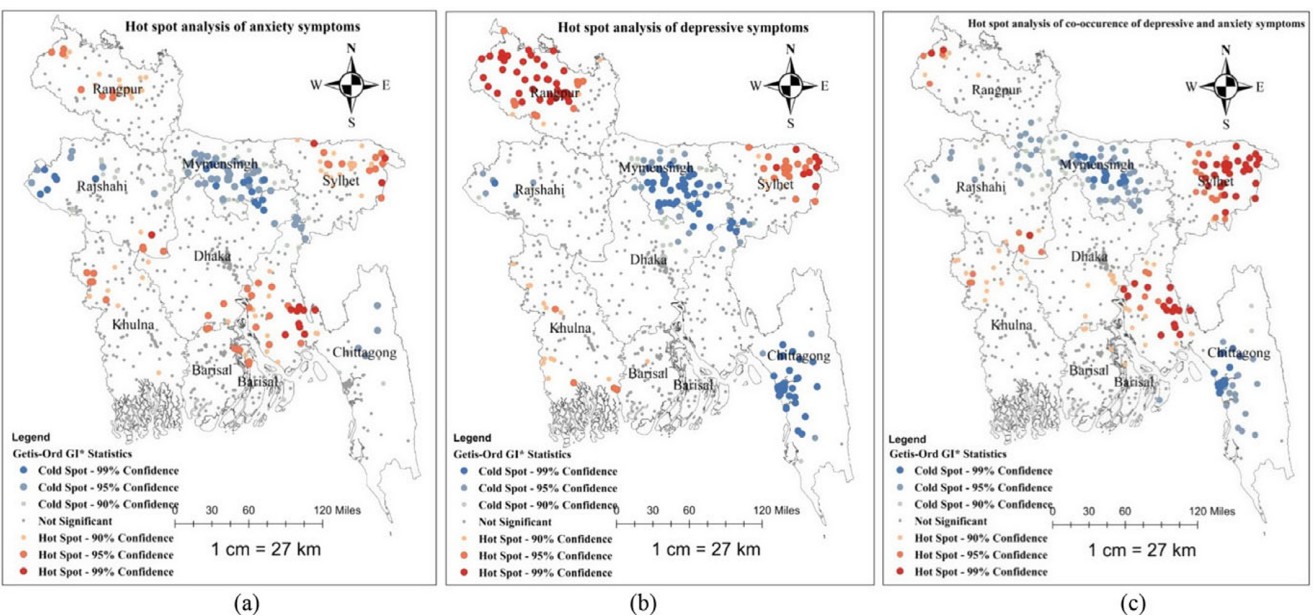

**Figure 2.** (a) Result for the hot spot analysis of anxiety symptoms. (b) Result for the hot spot analysis of depressive symptoms. (c) Result for the hot spot analysis of CAD symptoms. All maps were generated using ArcGIS v10.8 (https://www.arcgis.com/index.html) with data from the BDHS 2022 survey, utilising the base shapefile of Bangladesh from a freely available online source: https://data.humdata.org/dataset/cod-ab-bgd?

(M. Rahman et al., 2012; Wahid et al., 2024). Given the disproportionate IPV burden in LMICs, partner violence must be addressed using legal protection, community education and women's empowerment interventions to reduce the burden of mental health disorders (Bott et al., 2005).

Our results indicated a robust relationship between lower educational levels and greater probabilities of CAD symptoms in women. That is, women with less than a high school education had significantly greater odds of experiencing these symptoms compared to those with more than a high school education. This finding is in agreement with earlier studies that have suggested higher educational achievement is a protective factor against the onset of common mental disorders (Bjelland et al., 2008; Cheah et al., 2020). Education is thought to improve mental health by providing better access to knowledge, encouraging better coping mechanisms and providing a sense of mastery and autonomy, all of which are essential for managing psychological distress (Halme et al., 2023). Similar patterns have been observed among European and Southeast Asian populations, where better-educated women have shown lower levels of anxiety and depressive symptoms (Bjelland et al., 2008; Cheah et al., 2020; Halme et al., 2023). Furthermore, socio-economic disparities, such as lower wealth index, have been linked to heightened psychological distress, while cultural stigma surrounding mental illness and emerging climate stressors (e.g., flooding and heatwaves) can exacerbate the burden of anxiety and depression among women in low- and middle-income settings (Lund et al., 2010; Hayes et al., 2018). However, some studies indicate that while the education of a woman herself is not necessarily directly correlated with depression, that of her husband might be predictive of greater psychological susceptibility (Ali et al., 2002; Jagannathan et al., 2010). Therefore, improving education levels among women and their husbands as well might become an effective tool in reducing the feeling of anxiety and depression in LMICs (Chazelle et al., 2011).

Sexual and reproductive health remains a cultural taboo in Bangladesh, especially regarding information and services, leaving people vulnerable to health risks (Ainul et al., 2017). About half of adolescent girls in Bangladesh feel fearful and shy about puberty issues; as a result, they are reluctant to disclose sexual and reproductive health problems (Alam et al., 2024). Additionally, Das and Malakar (2022)'s study in 2020 showed that 31% of males experience erectile dysfunction during the survey period. Our analysis showed that sexually inactive women were far more likely to be experiencing both depressive and anxiety symptoms than sexually active ones. It is supported by previous work showing sexual satisfaction to be of paramount importance for emotional well-being. Without sexual intimacy, particularly in marital life, it is linked with loneliness, emotional alienation and decreased quality of life (Flynn et al., 2016; Khakkar and Kazemi, 2023). Without sexual intimacy in societies like Bangladesh, where emotional intimacy in marriage is valued very much, even this might augment women's psychological distress, especially when the emotional needs of women are unfulfilled (Koly et al., 2022). Sexual behaviour also has a positive influence on mental health by enhancing self-esteem, emotional attachment and stress regulation (Klein and Gorzalka, 2009). Sexual health, however, remains systematically overlooked in mental health care. Adding sexual pleasure discussions and marital communication to mental health treatment may be the first step toward the promotion of general well-being in women of reproductive age.

Our findings revealed that individuals whose husbands were unemployed had more than double the risk of experiencing both anxiety and depression compared to their peers whose husbands were employed in the service sector. This accords with earlier studies proving that unemployment bears implications not just for a person but even for his or her partner's mental health, increasing risks of emotional strain and vulnerability to mental distress (Bubonya et al., 2017; Paul and Moser, 2009). In Bangladesh's cultural and scarce-resource settings, where men are often the sole breadwinners, male unemployment can disrupt household stability and increase women's caregiving burdens, limiting their access to help and mental health care

(M. A. Islam et al., 2016). Moreover, households where the husband is unemployed and belongs to a lower wealth index face heightened psychological stress, as financial insecurity intersects with cultural stigma around male joblessness and climate-related livelihood disruptions, compounding women's vulnerability to anxiety and depression (Bakebillah et al., 2024).

The spatial analysis and logistics regression analysis revealed that, among the eight divisions, the significant hotspots for the anxiety symptoms among currently married reproductive-aged women were identified in the Rangpur, Sylhet and Chittagong regions. Besides, Rangpur and Sylhet regions were identified as hotspot areas for depressive symptoms, and Sylhet and Chittagong regions were identified as hotspot areas for CAD symptoms among currently married women. In contrast, Mymensingh and Rajshahi regions were pinpointed as cold spot areas for the anxiety symptoms, while Chittagong, Mymensingh and Rajshahi regions were noted as cold spot areas for the depressive symptoms. Chittagong, Mymensingh, Rajshahi and Rangpur were identified as cold spot areas for the CAD symptoms. A previous study in Bangladesh showed a higher prevalence of depression among university entrance test-takers in Takurgaon, Nilphamari, Lalmonirhat and other western districts (Al-Mamun et al., 2024).

Additionally, gender-based anxiety was high in the Rajshahi and Lalmonirhat districts (Al-Mamun et al., 2024). Northern regions (such as Rangpur) are significantly impacted by trauma and stress for weaker to natural disasters (such as droughts and flooding), which are causative of a more serious problem of mental health issues (Stanke et al., 2012). Sylhet and Rangpur can lead to feelings of despair and hopelessness, which increase the risk of anxiety and depressive symptoms (Boreham and Schutte, 2023). Almost 40% and 47.2% of the populations of Mymensingh and Rangpur regions, respectively, live in the poorest wealth quintile (Bangladesh Bureau of Statistics [BBS], 2023). Moreover, both Mymensingh and Rangpur regions report lower female literacy rates, with 65% and 68%, respectively, compared to the national average of 73% (BBS, 2022). Besides, currently married women of the middle region in Bangladesh tend to have better access to employment opportunities, while the lack of such opportunities exists in other regions. A meta-analysis shows that a greater purpose in life was significantly related to lower levels of depression (Boreham and Schutte, 2023).

One-Stop Crisis Centers are established by the Ministry of Women and Child Welfare, providing psychological counselling to women survivors of trauma and violence (Ministry of Health and Family Welfare, 2022). We suggest expanding and enhancing these facilities in Sylhet, Chittagong and Rangpur divisions. At the community level, strengthening awareness and peer-support networks can empower women to seek help early, while individual-level interventions should focus on improving mental health literacy and resilience. Besides, the government should expand community-based awareness programmes to break down the stigma about mental health and digital mental health services to increase accessibility in remote regions of Bangladesh.

### *Strengths and limitations*

This study has several relevant strengths. To our knowledge, it is among the first to investigate spatial patterns of anxiety, depressive symptoms and their co-occurrence among currently married reproductive-aged women in Bangladesh using the recently released BDHS 2022 data. The BDHS is a nationally representative population-based survey with a sound sampling design and a high response rate, enhancing our findings' reliability and comparability, thereby being generalizable across settings with similar socio-economic contexts. Our focus on currently married reproductive-aged women adds specificity and policy relevance to the anxiety, depressive symptoms and their co-occurrence in Bangladesh.

This study also has some limitations. BDHS relies on self-report data that is subject to recall bias and social desirability bias, which may distort the accuracy of self-reported symptoms of anxiety, depressive symptoms and their co-occurrence. Additionally, we considered only currently married women of reproductive age for this study, excluding missing data of the outcome variable. The exclusion of divorced, widowed or separated women from the analysis may have led to an underestimation of the overall prevalence, as these groups are often at higher risk of experiencing anxiety and depressive symptoms. Additionally, a substantial proportion of cases were excluded due to missing data, which may have introduced selection bias and affected the representativeness of the findings. Due to its cross-sectional design, the study cannot make causal inferences between anxiety, depressive symptoms and their co-occurrence and their associated factors. Furthermore, the analysis could not control for several important psychosocial and clinical variables that may influence anxiety, depressive symptoms and their co-occurrence. These include factors such as a history of psychiatric disorders, urinary incontinence, diabetes, trauma exposure, substance use, comorbidities, poverty and disaster exposure. In addition, the BDHS dataset did not provide information on non-pharmacologic interventions (e.g., cognitive-behavioural therapy and counselling), limiting insight into treatment and support practices. The unmeasured confounders and the absence of robustness tests may have influenced the observed associations. Additionally, DHS cluster coordinates are intentionally displaced slightly to protect confidentiality, which may reduce hotspot precision and introduce potential ecological fallacy in spatial interpretation. These limitations should be considered when interpreting the findings.

### Conclusion

In conclusion, the findings of this study highlight the high mental health burden of anxiety, depressive symptoms and their co-occurrence among currently married reproductive-age women in Bangladesh, with notable regional differences. Our research identified key factors associated with the presence of anxiety and depressive symptoms in these women, including a higher wealth index, experiencing frequent spousal violence, lower high school completion, sexual inactivity and having unemployed husbands. Rangpur, Sylhet and Chittagong were identified as hotspot areas for anxiety symptoms; Rangpur and Sylhet for depressive symptoms; and Sylhet and Chittagong for CAD symptoms among currently married women of reproductive age. There is an urgent need to develop and implement targeted interventions to address regional disparities, considering social, cultural and economic contexts. Community-level mental health interventions should be tailored to vulnerable groups. Culturally appropriate intervention models piloted in the region, such as One-Stop Crisis Centers, community-based mental health worker programmes and digital outreach initiatives, provide practical frameworks for implementing targeted responses aligned with local contexts. Future research should adopt longitudinal and mixed-methods designs to capture temporal changes and a deeper contextual understanding of mental health determinants. Studies should evaluate the effectiveness of interventions, assess existing mental health programmes and integrate

spatial data with health service mapping to identify service gaps. Additionally, research should include diverse and vulnerable populations, such as unmarried women, adolescents and the elderly, to inform equitable and evidence-based mental health system strengthening in Bangladesh.

**Open peer review.** To view the open peer review materials for this article, please visit http://doi.org/10.1017/gmh.2025.10121.

**Supplementary material.** The supplementary material for this article can be found at http://doi.org/10.1017/gmh.2025.10121.

**Data availability statement.** Data are publicly available at this link: https://dhsprogram.com/data/

**Acknowledgements.** The authors would like to express their gratitude to the DHS Program for providing public access to the 2022 Bangladesh Demographic and Health Survey data for analysis. The authors would also like to acknowledge the contributions of the DHS country coordinators (NIPORT), as well as other relevant organisations, such as ICF and USAID, and the survey participants.

**Author contribution.** M.A.H. had full access to all of the data in the study and takes responsibility for the integrity of the data and the accuracy of the data analysis. M.A.H. and S.K. created the concept and designed the study. M.A.H. and M.Y.A. validated the data. M.A.H. and S.K. performed the statistical analysis. M.A.H. and M.Y.A. worked on the original draft of the manuscript. M.A.H., M.Y.A. and S.K. reviewed and edited the manuscript critically. All authors read and approved the final manuscript.

**Financial support.** The authors have declared no funding support for this study.

**Competing interests.** The authors declare none.

**Ethics statement.** This present study used secondary sources of data and therefore did not require ethics approval. However, the BDHS data 2022 was reviewed and approved by the Ministry of Health and Family Welfare. Informed consent was obtained by the original BDHS survey teams, and we accessed the fully de-identified dataset after obtaining approval through the DHS Programme's data request system.

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
