## [Reviewer Report]

Comment

This manuscript presents a well-conceived and methodologically rigorous study investigating the spatial distribution and associated determinants of anxiety, depressive symptoms, and their co-occurrence (CAD) among married, reproductive-aged women in Bangladesh using nationally representative data from the Bangladesh Demographic and Health Survey (BDHS) 2022. The paper’s strength lies in its integration of spatial analysis (Moran’s I, Getis-Ord Gi*) with multivariable logistic regression, offering novel insight into the geographic disparities and psychosocial determinants of women’s mental health in a low- and middle-income country context. The study is well-aligned with the aims and scope of Cambridge Prisms: Global Mental Health.

I commend the authors for their robust statistical approach, clear reporting of prevalence estimates and odds ratios, and policy-oriented discussion. However, several areas require clarification or refinement prior to publication. I recommend minor revisions.

Major Points:

1. Terminology of “Wife Beating”:

While the authors operationalize domestic violence via PCA-derived indicators, the use of the term “wife beating” is outdated and inappropriate for scholarly communication. I recommend replacing all occurrences with “intimate partner violence (IPV)” for consistency with global health terminology.

2. Conceptual Framework and Theoretical Framing:

The manuscript would benefit from a brief conceptual framing—such as the social determinants of health or an ecological model of mental health—to situate the analysis and findings in a broader theoretical context. This would also assist in interpreting interaction between socioeconomic, demographic, and geographic factors.

3. Measurement Limitations of Sexual Inactivity:

The operationalization of “sexual inactivity” as absence of intercourse in the past month lacks granularity. Please consider discussing its potential conflation with relational distress or health conditions, and its cultural interpretation in Bangladesh.

4. Spatial Analysis Limitations:

Given the use of GPS-displaced DHS cluster data, the manuscript should acknowledge the implications of spatial displacement for hotspot precision and the risk of ecological fallacy in spatial interpretation.

5. Discussion of Intervention Models:

The conclusion outlines the need for targeted interventions. It would strengthen the paper to briefly reference or suggest culturally appropriate models (e.g., One-Stop Crisis Centers, community mental health workers, digital outreach) already piloted in the region.

Minor Points:

1. Abstract: Please spell out the acronym “CAD” before first use.

2. Introduction: The prevalence data from other South Asian countries (e.g., Nepal, India) should be presented in one consolidated paragraph for better flow.

3. Figures & Tables: The supplementary figures are referenced in the text but not included in the proof. Ensure their inclusion for review.

4. Tables 1–3: Consider color highlighting or bolding statistically significant AORs to enhance readability.

Conclusion:

This is a valuable and timely contribution to the literature on gendered mental health disparities in LMICs. The integration of spatial epidemiology with nationally representative survey data adds important insights for targeted mental health intervention and policy. Subject to minor revisions addressing the points outlined above, I recommend the manuscript for publication.

Recommendation: Minor Revision

---

## [Reviewer Report]

The study is valuable in highlighting spatial disparities in Bangladeshi women’s mental health, but its weaknesses lie in title precision, limited causal inference, reliance on self-report, exclusion of high-risk groups, and lack of deeper contextual/theoretical integration. Its contribution is important for geographic targeting of interventions, though it advances knowledge only incrementally.

General comments

Quality of paper

- Cross-sectional design: prevents causal inference between determinants and outcomes.

- Self-report measures: vulnerable to recall bias and social desirability bias, which can underreport sensitive issues like domestic violence or mental health stigma.

- Excluded populations: divorced, widowed, or separated women were excluded, despite being potentially at higher risk of poor mental health.

- Uncontrolled confounders: No adjustment for important psychosocial and clinical factors (e.g., prior psychiatric history, trauma, comorbid health conditions, or treatment history).

- Limited treatment context: No data on availability or use of mental health services, non-pharmacological interventions, or cultural coping practices.

Abstract

The abstract states “Bangladeshi women” but the study actually focuses only on married reproductive-aged women (15–49 years), which is not made explicit. This may mislead readers about generalizability.

The abstract reports associations but does not explicitly mention the cross-sectional design, which limits causal interpretation.

Mental health outcomes (GAD-7, PHQ-9) are self-reported, and the abstract does not acknowledge possible recall bias or social desirability bias.

The abstract does not note the absence of clinical or psychosocial variables (e.g., psychiatric history, comorbidities, trauma, health service use) that may affect results.

The abstract omits the fact that divorced, widowed, and separated women were excluded, even though they may have higher risk of poor mental health.

The abstract concludes that findings can “help design site-specific programs”, but does not specify concrete intervention directions, leaving the contribution vague.

Introduction

The introduction is informative but lacks theoretical grounding, precise justification for scope, and clarity on novelty. It frames the problem in epidemiological terms but misses an opportunity to link findings to policy action, cultural context, and conceptual models.

The introduction starts with global prevalence and burden of mental illness but does not sufficiently situate the unique cultural, social, and health system context of Bangladesh. As a result, the transition from global to local is somewhat abrupt and lacks depth.

The context cites multiple prevalence studies (GBD, IHME, WHO) but does not adequately critique inconsistencies across data sources (e.g., global modeled vs. local survey data). This weakens the case for why new spatial analysis is necessary.

The study restricts to married reproductive-aged women (15–49 years), yet the introduction does not explain why unmarried, widowed, or divorced women (often more vulnerable) were excluded.

The problem framing is focused on socio-demographics (education, wealth, IPV), but neglects broader determinants such as cultural stigma, migration, environmental stressors, or healthcare accessibility.

The introduction notes disparities in prevalence but does not sufficiently demonstrate how identifying spatial patterns directly contributes to policy or programmatic interventions. Justification remains descriptive rather than solution-oriented.

The study is framed descriptively without anchoring in a clear theoretical model (e.g., social determinants of health, ecological model, stress-vulnerability framework). This limits the ability to interpret mechanisms linking determinants and outcomes.

The stated aim is to “assess spatial patterns and identify factors”, but this is generic and does not articulate specific hypotheses or measurable research questions.

Although the study claims to be the first spatial analysis of anxiety and depression among Bangladeshi women, the introduction does not explicitly differentiate it from prior BDHS-based studies, which already examined prevalence and determinants.

Methods

The Methods section is statistically sound but limited by design (cross-sectional, self-report), exclusions (narrow sample, omitted vulnerable groups), and analytical constraints (confounders, GPS displacement, lack of robustness tests). These weaknesses reduce causal clarity and external validity.

The study relies on BDHS 2022, which is cross-sectional; thus, causal inference between determinants and mental health outcomes cannot be established.

Only married reproductive-aged women (15–49 years) were included, while divorced, widowed, and unmarried women (potentially at higher risk) were excluded without strong methodological justification.

Anxiety (GAD-7) and depression (PHQ-9) were assessed through self-reported tools, which may be influenced by recall bias, underreporting, or cultural stigma.

The classification of moderate to severe anxiety/depression at ≥10 on GAD-7/PHQ-9 may omit women with mild but clinically relevant symptoms, underestimating the burden.

Important psychosocial and clinical variables were not available or controlled for (e.g., psychiatric history, substance use, trauma exposure, comorbidities, or access to mental health care).

Exclusion of women with sick children (e.g., diarrhea, fever, cough) may have biased results, as maternal mental health is often linked to child health.

GIS analysis (Moran’s I, hotspot detection) used cluster-level prevalence but did not consider spatial autocorrelation of confounding variables (e.g., poverty, disaster exposure).

Also, BDHS GPS data are randomly displaced for confidentiality, which can reduce precision of hotspot mapping.

Logistic regression only included variables significant in bivariate analysis, which risks omitting potential confounders.

No robustness checks (e.g., sensitivity analysis, multilevel modelling, or cross-validation) were performed.

Results

The Results section provides detailed prevalence estimates and regression findings but is weakened by underreporting of mild cases, limited causal caution, exclusion of high-risk groups, insufficient exploration of hotspot context, and lack of robustness tests. It risks being statistically strong but policy-weak.

The results only report moderate to severe anxiety and depressive symptoms, omitting mild cases. This underestimates the true burden of mental health distress in the population.

Findings are presented in a way that may suggest causality (e.g., “women with X were more likely to have depression”), but the cross-sectional design prevents establishing directionality.

The exclusion of divorced, widowed, or separated women means the reported prevalence may not reflect the highest-risk populations.

While hotspots (Rangpur, Sylhet, Chittagong) are identified, the results do not adequately explore why these regions are high-burden (e.g., disaster vulnerability, poverty, cultural stigma).

The results rely on BDHS cluster GPS data, which are deliberately displaced for confidentiality. This reduces the accuracy of spatial clustering and hotspot identification, but this limitation is not highlighted in the results section.

Only logistic regression outputs are presented. No sensitivity analysis, multilevel modelling, or stratification (e.g., rural vs. urban differences) is reported, which limits confidence in the robustness of associations.

Large tables of odds ratios are presented without sufficient narrative synthesis, making it harder for readers to grasp which determinants are most meaningful for intervention.

The results highlight regional disparities and socio-demographic determinants but do not connect them to practical implications in the results narrative (this is deferred to the discussion).

Discussion

The Discussion is weakened by being too descriptive, lacking theoretical depth, offering broad rather than actionable recommendations, and giving only generic suggestions for future research. While the limitations acknowledged are valid, important issues like data displacement, excluded populations, and methodological improvements (e.g., longitudinal or mixed-methods approaches) are overlooked.

The discussion mostly restates prevalence and associations rather than critically interrogating why certain regions and groups are at higher risk. Broader structural factors (healthcare system gaps, cultural stigma, migration, climate stressors) are mentioned only briefly or not at all.

Findings are not interpreted through a clear theoretical framework (e.g., social determinants of health, ecological systems theory). This weakens the contribution to conceptual understanding, leaving results largely descriptive.

While the discussion compares prevalence with some South Asian studies, it does not engage with conflicting or outlier evidence (e.g., why Bangladesh estimates differ from other LMICs despite similar contexts).

Practical Implications

The section calls for “site-specific programs” and “gender-sensitive interventions” but does not provide concrete, evidence-based strategies (e.g., integration of community-based counselling, expansion of One-Stop Crisis Centres, mobile mental health services).

Practical implications do not distinguish between individual-level, community-level, and system-level responses. This limits translation into actionable policy design.

Limitations and Future Studies

The authors mention self-report bias, cross-sectional design, and missing clinical variables.

However, they do not discuss the implications of GPS displacement in BDHS, which weakens spatial hotspot accuracy, nor the exclusion of divorced/widowed women (a high-risk group).

The paper only hints that “future studies are needed” without specifying which methods (e.g., longitudinal designs, qualitative studies, mixed methods, intervention trials) or which populations (e.g., unmarried women, adolescents, elderly).

Future research directions do not address testing of interventions, evaluation of mental health programs, or integration of spatial data with health service mapping. This reduces the paper’s utility for guiding mental health system strengthening.

Conclusion

The Conclusion is weakened by overgeneralisation, repetition, vague policy guidance, and lack of forward-looking research directions. It misses the opportunity to deliver a sharper synthesis that positions the study’s findings as a springboard for both policy action and scholarly advancement.

The conclusion largely re-summarises prevalence rates, hotspots, and determinants already presented in the results, instead of synthesising key insights or providing a stronger conceptual take-away.

The conclusion frames results as applying to “Bangladeshi women” broadly, even though the study only covered married reproductive-aged women. This risks overstating generalisability.

Despite acknowledging cross-sectional design elsewhere, the conclusion sometimes implies cause–effect relationships between determinants (e.g., IPV, education, unemployment) and mental health outcomes, which cannot be confirmed.

---

## [Reviewer Report]

Summary of Highlights. Please the full details in what i forwarded to the Editor.

Title of the manuscript:

Dear researcher(s), you are addressing an important gap. Your paper is fairly-written and it has some important results, and if you edit your paper it can be much more effective. Here are some humble suggestions to improve the paper, I would do the following to strengthen the paper. You need to edit some areas in the manuscript and then the Journal editor would have to send the edited copy to me after your corrections, for me to see it before the paper is published. You could increase the effect of your paper with some more recent studies suggested if available.

Main points:

1. Title: “Spatial patterns and determinants of anxiety, depressive symptoms and their co-occurrence among Bangladeshi women”

a. It is good as it correlates with content of the manuscript, but see my full comment.

2. Abstract and keywords: fair and comprehensive:

3. Overall language:

- The language is clear and fairly-written. However, you have to re-write/summarize some parts as i highlighted in the manuscript eg exclusion criteria etc.

4. Length of paragraph :

a. Fair and you can check the paper and make sure every paragraph is not more than 5 sentences. The best is to stick with 3 to 5 sentences.

5. Introduction: Good

6. Thoroughness of the literature review: can be supported with some recent studies, yet you do not have to do it if there are no recent articles. This will increase the effect of the paper and the journal.

7. Research design:

a. Is good.

8. Clearly providing research questions and/or purpose: is good, as it has been indicated in the aim, but some journals may require hypothoses

9. Choice of research method: Is good.

10. Appropriateness of procedures chosen for data collection and analysis: Its well-written

11. Relevance of data obtained in view of the purpose of the research: Well-written

12. Discussion of the results and their significance: Well-written

13. Soundness of conclusions in relation to data presented: Conclusion is well written.

14. Limitation: Well-written.

15. Implication/Impact Statement:

a. Its well written.

b. I would strongly suggest you summarize this section.

16. Format of referencing: You have to follow what the journal indicates.

17. Figure/tables: well represented

18. In-text reference: well written

19. References:

- Please use the following link to include all available doi numbers https://doi.crossref.org/simpleTextQuery simply include your reference one or more than one at a time and submit it. Then you should get all doi numbers if a manuscript has it.

I am looking forward to seeing the corrected manuscript before the paper can be published.

---

## [Reviewer Report]

Thank you very much for the opportunity to review this paper submitted for publication. In summary, the authors present findings on the prevalence, determinants and spatial patterns of anxiety, depression and the co-occurrence of both anxiety and depression among reproductive age women, using data from the Bangladesh Demographic and Health Survey (BDHS) 2022. The authors have addressed a significant public health issue, reporting new and interesting findings within the context of Bangladesh. However, there are methodological concerns that impede the quality of the work presented here.

Comments

Page 5 line 40: This study aims to assess the spatial patterns of anxiety, depressive symptoms, and their co-occurrence, and identify the factors associated with these mental health outcomes among reproductive-aged women in Bangladesh.

Page 6, exclusion criteria: …For this reason, 16640 women were omitted from the analysis out of 30078 reproductive-aged women, based on the following criteria: 10,090 with missing data;

2336 not living with their husbands at the time of the survey; 174 separated, 532 widowed, and

260 divorced. Additionally, 393, 2242, and 613 children were affected by diarrhea, fever, and

cough, respectively, in the last two weeks.

Results: The results presented throughout are based on reproductive-age married women

1) It is not clear what study population the article is based on, since there is a considerable disagreement regarding the study population mentioned in different parts of the manuscript. Reproductive-age women vs reproductive-age married women

2) The rationale for excluding different high-risk groups from the analysis is not clear. Since the study aimed to estimate the prevalence of anxiety, depression, their co-occurrence, as well as their determinants, excluding persons expected to be at higher risk will underestimate the prevalence.

3) The study population is reproductive-age women according to the aim, and the results are reported for reproductive-age married women. What about single women who are of reproductive age?

This important weakness in the study was not acknowledged in the limitations.

4) The authors failed to assess possible bias introduced by missing data since 1/3 of the sample was deleted due to missing data. The possibility of bias due to missing data was not mentioned as a limitation.

5) I recommend that authors substitute terms like “those whose husbands beat them” and “wife beating” for “exposure to intimate partner violence” or a similar term.

6) Page 6, line 41 and line 53 should read “women’s responses to each item…….”

7) Kindly rename some of the variables as follows: women’s educational level to educational level, and women’s age to just age. There is no need for the prefix “women” since the sample is already restricted to women.

8) Page 15, line 51: The explanation is not true for the poorest categories. The trend is also irregular as the level of socioeconomic disadvantage increases

---

## [Editor Report]

Dear Dr. Satyajit Kundu, 

Manuscript ID GMH-2025-0212 entitled “Spatial patterns and determinants of anxiety, depressive symptoms and their co-occurrence among Bangladeshi women,” which you submitted to Cambridge Prisms: Global Mental Health, has been reviewed. The comments of the reviewer(s) and editor(s) are included at the bottom of this letter. 

The manuscript is not acceptable for publication in its current form. However, I invite you to revise the manuscript in accordance with the reviewers' and editor’s comments below and submit a revision. 

When submitting your revised manuscript, you will respond to the comments made by the reviewer(s). In order to expedite the processing of the revised manuscript, please be as specific as possible in your response to the reviewer(s).

Once again, thank you for submitting your manuscript to Cambridge Prisms: Global Mental Health, and I look forward to receiving your revision.

---

## [Reviewer Report]

All comments are well-suited for revision and meet the quality standards of the journal. Good luck!

---

## [Reviewer Report]

1. Line 51, delete the word “highly”

2. Line 17 - 22, Pls use the standardized definitions of anxiety and depressive disorders as enshrine by the accepted manuals for example ICD11 Or DSM5!

For example in the definition of depression you stated, the “loss of interest” must be in previously pleasurable activities.

“Anxiety defined by inappropriate fearfulness, worry, and impairment of behaviour, while depressive disorders are expressed by persistent low mood, loss of interest, and impairment of function (Ferrari et al., 2013; WHO, 2019).”

Finally i have enjoyed reading the paper, i look forward to the paper being published after effecting the minor corrections on the manuscript.

---

## [Reviewer Report]

Comments to the authors

This manuscript addresses an important and underexplored topic: spatial patterns and determinants of anxiety, depressive symptoms, and their co-occurrence among currently married women of reproductive age in Bangladesh using BDHS 2022 data. The combination of validated screening tools (PHQ-9, GAD-7), multivariable logistic regression, and spatial autocorrelation / hot spot analysis provides a strong empirical basis for the findings and clear implications for geographically targeted mental health policy and programming.

The paper is generally well structured, the methods are appropriate, and the results are clearly aligned with the stated objectives. I support publication after the authors address the following minor issues, which mainly concern clarity of writing, consistency of terminology, and a few methodological clarifications.

1. Language and style editing throughout

There are several places where the English needs light editing for clarity and to remove remnants of tracked changes or duplicated phrases. For example (non-exhaustive):

• Abstract and early pages: phrases such as “currently married women of, reproductive -aged women in Bangladesh”, “cross-sectional design data … which used a cross-sectional design”, and “co-occurringCAD symptoms” should be cleaned and simplified.

• Results sections: some sentences appear to have overlapping edits, e.g. “currently married women women who aged 25–34 years”, “women who beat their husbands frequently” (which likely intends “women who are frequently beaten by their husbands / exposed to high IPV”), and similar constructions.

• Strengths and limitations: some sentences are partially duplicated (e.g. limitations relating to self-report, missing data, unmeasured confounders, and non-pharmacologic interventions appear twice with slightly different wording).

A careful language edit (or professional copy-edit, if available) will substantially improve readability without changing the content.

2. Consistent terminology for the population and outcomes

Please harmonise terminology across the manuscript:

• Use one consistent phrase for the target group, e.g. “currently married women of reproductive age” rather than alternating with “reproductive-aged women,” “reproductivecurrently married reproductive-aged women,” etc.

• Similarly, standardise the terminology for the combined outcome: choose either “co-occurrence of anxiety and depressive symptoms” or “co-occurring anxiety and depressive (CAD) symptoms”, define the acronym once, and then use it consistently in the Abstract, Methods, Results, Figures, and Discussion.

• The title appears in slightly different forms in the document (e.g. “…in Bangladesh” vs “…in Bangladeshi women”). Please select a single final version and make sure it matches the journal’s title page requirements.

3. Naming of “wife beating” variable (intimate partner violence)

The manuscript currently uses phrases such as “wife beatingIntimate partner violence (IPV)” and “women who are highly beaten by their husbands.” For an international mental health and public-health audience, it would be preferable to:

• Consistently use “intimate partner violence (IPV)” as the main term,

• Avoid the phrase “wife beating” except perhaps in brackets when explaining the original BDHS question wording, and

• Ensure that the direction of violence is unambiguous (e.g. “women exposed to high levels of IPV from their husbands”).

Please also ensure that tables, figure legends, and the narrative use the same terminology and categories for IPV.

4. Exclusion criteria and selection bias – clarify and align with limitations

The Methods section describing the exclusion criteria is important but currently somewhat confusing. You state that divorced/widowed/separated women and mothers of children with recent diarrhoea, fever or cough were excluded, and later acknowledge that such exclusions may lead to underestimation of mental health problems.

I suggest:

• Presenting a concise and transparent description of all exclusions in one place (e.g. in “Study design and participants” or in a short Figure / flow diagram showing the number of women excluded for each reason and the final weighted sample size.

• Briefly justifying why mothers of recently ill children were excluded (this is not a common restriction in mental-health surveys and may be questioned by readers), and explicitly acknowledging in the Limitations section that these exclusions could bias prevalence estimates downward and affect generalisability.

• Ensuring that the numbers of excluded participants add up clearly and that “weighted” vs “unweighted” counts are clearly distinguished.

These clarifications can be made in text and/or a simple flow figure and do not require re-analysis.

5. Interpretation of logistic regression results

The logistic regression analyses are appropriate and well presented; however, some wording occasionally implies stronger causal language than is warranted by the cross-sectional design. Examples include phrases such as “determinants” or “predictor factors” followed by causal verbs.

Minor edits are recommended to maintain a cautious interpretive tone, such as:

• Prefer “factors associated with anxiety/depressive/CAD symptoms” or “higher odds of symptoms” instead of implying causation.

• When discussing results in the Discussion, continue to emphasise that these are associations identified in cross-sectional data.

You already note in the Limitations that causal inference is not possible; aligning the wording throughout will strengthen the methodological rigour.

6. Tables and alignment with text

Tables 1–3 are informative, but a few small improvements would enhance clarity:

• Check that reference categories are consistently labelled in each table (e.g. “Reference” for the baseline category for age, region, education, wealth, etc.).

• Ensure the variable labels match those in the text (e.g. “Women’s education level,” “Husband’s occupation,” “Intimate partner violence (IPV)” rather than “wife beating”).

• If space allows, consider adding a short table footnote clarifying all acronyms (e.g. CAD, IPV, BMI, UOR, AOR, CI).

Also verify that the prevalence patterns described in the text (for example, which divisions are hotspots for anxiety vs depression vs CAD) exactly match the patterns shown in the tables and maps.

7. Spatial analysis description and figures

The description of Global Moran’s I and Getis-Ord Gi* hot spot analysis is clear and appropriate. A few minor clarifications would help readers less familiar with spatial statistics:

• In the Methods – Spatial autocorrelation and hot spot analysis, consider adding one brief sentence explaining what a positive Moran’s I in this context means in practical terms (e.g. clustering of high or low prevalence across neighbouring clusters).

• In the figure captions for the spatial maps (global autocorrelation and hot spot maps), clearly indicate the colour scale and what “hotspots” and “cold spots” represent (e.g. “clusters of high/low prevalence significant at p < 0.05”).

These are minor editorial changes and do not alter the analysis.

8. Strengths and limitations – streamline and avoid repetition

The Strengths and limitations section is a valuable part of the paper. At present there is some redundancy (e.g. self-report bias, cross-sectional design, missing data and unmeasured confounders are mentioned more than once with similar wording). I recommend:

• Merging overlapping sentences into a concise set of 3–5 clearly differentiated limitations,

• Keeping the strongest points (e.g. self-report bias and social desirability, exclusion of divorced/widowed/separated women and missing data, cross-sectional design, lack of key psychosocial/clinical variables, and displaced GPS coordinates), and

• Ensuring that each limitation is linked to a plausible direction of bias (e.g. likely underestimation of prevalence, possible attenuation or inflation of associations).

This will make the section more focused and easier to follow.

9. Ethical considerations and data use statement

If not already present in sections not visible in this version, please ensure that there is a brief Ethics or Data access statement that:

• Notes that BDHS 2022 data use was approved by the relevant authorities,

• States that informed consent was obtained by the original survey teams, and

• Indicates that the authors received permission to use the de-identified dataset (e.g. via the DHS Program data request process).

This is standard for secondary analyses of DHS data and is likely already included; if so, only ensure it is placed where the journal expects it.

---

## [Editor Report]

Dear Dr Satyajit, Kundu 

We thank you for submitting your revisions. The first round of the peer review process of your manuscript has now been completed, and we have reached a decision regarding your submission.

At present, your manuscript requires minor revisions to address the concerns of

the reviewers. Their comments are attached to the email and/or at the bottom

of this letter. 

Please include with your revised submission an itemised, point-by-point response to the reviewers that details the changes made. If you anticipate that you will be unable to meet this deadline, please notify the Editorial Office.

Reviewer 2 Comments

1. Line 51, delete the word “highly”

2. Line 17 - 22, Pls use the standardized definitions of anxiety and depressive disorders as enshrine by the accepted manuals for example ICD11 Or DSM5!

For example in the definition of depression you stated, the “loss of interest” must be in previously pleasurable activities.

“Anxiety defined by inappropriate fearfulness, worry, and impairment of behaviour, while depressive disorders are expressed by persistent low mood, loss of interest, and impairment of function (Ferrari et al., 2013; WHO, 2019).”

Finally i have enjoyed reading the paper, i look forward to the paper being published after effecting the minor corrections on the manuscript.

Reviewer 3 Comments

Comments to the authors

This manuscript addresses an important and underexplored topic: spatial patterns and determinants of anxiety, depressive symptoms, and their co-occurrence among currently married women of reproductive age in Bangladesh using BDHS 2022 data. The combination of validated screening tools (PHQ-9, GAD-7), multivariable logistic regression, and spatial autocorrelation / hot spot analysis provides a strong empirical basis for the findings and clear implications for geographically targeted mental health policy and programming.

The paper is generally well structured, the methods are appropriate, and the results are clearly aligned with the stated objectives. I support publication after the authors address the following minor issues, which mainly concern clarity of writing, consistency of terminology, and a few methodological clarifications.

1. Language and style editing throughout

There are several places where the English needs light editing for clarity and to remove remnants of tracked changes or duplicated phrases. For example (non-exhaustive):

• Abstract and early pages: phrases such as “currently married women of, reproductive -aged women in Bangladesh”, “cross-sectional design data … which used a cross-sectional design”, and “co-occurring CAD symptoms” should be cleaned and simplified.

• Results sections: some sentences appear to have overlapping edits, e.g. “currently married women women who aged 25–34 years”, “women who beat their husbands frequently” (which likely intends “women who are frequently beaten by their husbands / exposed to high IPV”), and similar constructions.

• Strengths and limitations: some sentences are partially duplicated (e.g. limitations relating to self-report, missing data, unmeasured confounders, and non-pharmacologic interventions appear twice with slightly different wording).

A careful language edit (or professional copy-edit, if available) will substantially improve readability without changing the content.

2. Consistent terminology for the population and outcomes

Please harmonise terminology across the manuscript:

• Use one consistent phrase for the target group, e.g. “currently married women of reproductive age” rather than alternating with “reproductive-aged women,” “reproductivecurrently married reproductive-aged women,” etc.

• Similarly, standardise the terminology for the combined outcome: choose either “co-occurrence of anxiety and depressive symptoms” or “co-occurring anxiety and depressive (CAD) symptoms”, define the acronym once, and then use it consistently in the Abstract, Methods, Results, Figures, and Discussion.

• The title appears in slightly different forms in the document (e.g. “…in Bangladesh” vs “…in Bangladeshi women”). Please select a single final version and make sure it matches the journal’s title page requirements.

3. Naming of “wife beating” variable (intimate partner violence)

The manuscript currently uses phrases such as “wife beatingIntimate partner violence (IPV)” and “women who are highly beaten by their husbands.” For an international mental health and public-health audience, it would be preferable to:

• Consistently use “intimate partner violence (IPV)” as the main term,

• Avoid the phrase “wife beating” except perhaps in brackets when explaining the original BDHS question wording, and

• Ensure that the direction of violence is unambiguous (e.g. “women exposed to high levels of IPV from their husbands”).

Please also ensure that tables, figure legends, and the narrative use the same terminology and categories for IPV.

4. Exclusion criteria and selection bias – clarify and align with limitations

The Methods section describing the exclusion criteria is important but currently somewhat confusing. You state that divorced/widowed/separated women and mothers of children with recent diarrhoea, fever or cough were excluded, and later acknowledge that such exclusions may lead to underestimation of mental health problems.

I suggest:

• Presenting a concise and transparent description of all exclusions in one place (e.g. in “Study design and participants” or in a short Figure / flow diagram showing the number of women excluded for each reason and the final weighted sample size.

• Briefly justifying why mothers of recently ill children were excluded (this is not a common restriction in mental-health surveys and may be questioned by readers), and explicitly acknowledging in the Limitations section that these exclusions could bias prevalence estimates downward and affect generalisability.

• Ensuring that the numbers of excluded participants add up clearly and that “weighted” vs “unweighted” counts are clearly distinguished.

These clarifications can be made in text and/or a simple flow figure and do not require re-analysis.

5. Interpretation of logistic regression results

The logistic regression analyses are appropriate and well presented; however, some wording occasionally implies stronger causal language than is warranted by the cross-sectional design. Examples include phrases such as “determinants” or “predictor factors” followed by causal verbs.

Minor edits are recommended to maintain a cautious interpretive tone, such as:

• Prefer “factors associated with anxiety/depressive/CAD symptoms” or “higher odds of symptoms” instead of implying causation.

• When discussing results in the Discussion, continue to emphasise that these are associations identified in cross-sectional data.

You already note in the Limitations that causal inference is not possible; aligning the wording throughout will strengthen the methodological rigour.

6. Tables and alignment with text

Tables 1–3 are informative, but a few small improvements would enhance clarity:

• Check that reference categories are consistently labelled in each table (e.g. “Reference” for the baseline category for age, region, education, wealth, etc.).

• Ensure the variable labels match those in the text (e.g. “Women’s education level,” “Husband’s occupation,” “Intimate partner violence (IPV)” rather than “wife beating”).

• If space allows, consider adding a short table footnote clarifying all acronyms (e.g. CAD, IPV, BMI, UOR, AOR, CI).

Also verify that the prevalence patterns described in the text (for example, which divisions are hotspots for anxiety vs depression vs CAD) exactly match the patterns shown in the tables and maps.

7. Spatial analysis description and figures

The description of Global Moran’s I and Getis-Ord Gi* hot spot analysis is clear and appropriate. A few minor clarifications would help readers less familiar with spatial statistics:

• In the Methods – Spatial autocorrelation and hot spot analysis, consider adding one brief sentence explaining what a positive Moran’s I in this context means in practical terms (e.g. clustering of high or low prevalence across neighbouring clusters).

• In the figure captions for the spatial maps (global autocorrelation and hot spot maps), clearly indicate the colour scale and what “hotspots” and “cold spots” represent (e.g. “clusters of high/low prevalence significant at p < 0.05”).

These are minor editorial changes and do not alter the analysis.

8. Strengths and limitations – streamline and avoid repetition

The Strengths and limitations section is a valuable part of the paper. At present there is some redundancy (e.g. self-report bias, cross-sectional design, missing data and unmeasured confounders are mentioned more than once with similar wording). I recommend:

• Merging overlapping sentences into a concise set of 3–5 clearly differentiated limitations,

• Keeping the strongest points (e.g. self-report bias and social desirability, exclusion of divorced/widowed/separated women and missing data, cross-sectional design, lack of key psychosocial/clinical variables, and displaced GPS coordinates), and

• Ensuring that each limitation is linked to a plausible direction of bias (e.g. likely underestimation of prevalence, possible attenuation or inflation of associations).

This will make the section more focused and easier to follow.

9. Ethical considerations and data use statement

If not already present in sections not visible in this version, please ensure that there is a brief Ethics or Data access statement that:

• Notes that BDHS 2022 data use was approved by the relevant authorities,

• States that informed consent was obtained by the original survey teams, and

• Indicates that the authors received permission to use the de-identified dataset (e.g. via the DHS Program data request process).

This is standard for secondary analyses of DHS data and is likely already included; if so, only ensure it is placed where the journal expects it.

We look forward to receiving the revised manuscript.

Thank you

Dr Limkile Mpofu

Handling editor

---

## [Editor Report]

Dear Dr Kundu,

Re: “Spatial patterns and determinants of anxiety, depressive symptoms, and their co-occurrence among currently married women of reproductive age in Bangladesh.”

We’re delighted to let you know that your manuscript has now been accepted for publication in Cambridge Prisms: Global Mental Health.

Dr Limkile Mpofu (PhD)

Department of Psychology (University of South Africa) 

Handling Editor (Cambridge Prisms: Global Mental Health).